# Machine Learning and Antibiotic Management

**DOI:** 10.3390/antibiotics11030304

**Published:** 2022-02-24

**Authors:** Riccardo Maviglia, Teresa Michi, Davide Passaro, Valeria Raggi, Maria Grazia Bocci, Edoardo Piervincenzi, Giovanna Mercurio, Monica Lucente, Rita Murri

**Affiliations:** 1Department of Emergency, Intensive Care Medicine and Anesthesia, Fondazione Policlinico Universitario Agostino Gemelli IRCCS, 00168 Rome, Italy; teresa.michi@gmail.com (T.M.); mariagraziabocci@gmail.com (M.G.B.); edoardo.piervincenzi@policlinicogemelli.it (E.P.); giovanna.mercurio@policlinicogemelli.it (G.M.); monica.lucente@gmail.com (M.L.); 2Department of Statistical Sciences, Università Sapienza, 00161 Rome, Italy; passaro.fk@gmail.com; 3Pediatric Cardiac Intensive Care Unit, Department of Cardiac Surgery, Cardiology and Heart Lung Transplant, Bambino Gesù Children’s Hospital, IRCCS, 00165 Rome, Italy; raggi.valeria@gmail.com; 4Infective Disease Department, Fondazione Policlinico Universitario Agostino Gemelli IRCCS-Università Cattolica Del Sacro Cuore, 00168 Rome, Italy; rita.murri@policlinicogemelli.it

**Keywords:** machine learning, fuzzy logic, clustering analysis, antibiotic therapy, intensive care unit

## Abstract

Machine learning and cluster analysis applied to the clinical setting of an intensive care unit can be a valuable aid for clinical management, especially with the increasing complexity of clinical monitoring. Providing a method to measure clinical experience, a proxy for that automatic *gestalt* evaluation that an experienced clinician sometimes effortlessly, but often only after long, hard consideration and consultation with colleagues, relies upon for decision making, is what we wanted to achieve with the application of machine learning to antibiotic therapy and clinical monitoring in the present work. This is a single-center retrospective analysis proposing methods for evaluation of vitals and antimicrobial therapy in intensive care patients. For each patient included in the present study, duration of antibiotic therapy, consecutive days of treatment and type and combination of antimicrobial agents have been assessed and considered as single unique daily record for analysis. Each parameter, composing a record was normalized using a fuzzy logic approach and assigned to five descriptive categories (fuzzy domain sub-sets ranging from “very low” to “very high”). Clustering of these normalized therapy records was performed, and each patient/day was considered to be a pertaining cluster. The same methodology was used for hourly bed-side monitoring. Changes in patient conditions (monitoring) can lead to a shift of clusters. This can provide an additional tool for assessing progress of complex patients. We used Fuzzy logic normalization to descriptive categories of parameters as a form nearer to human language than raw numbers.

## 1. Introduction

Machine learning (ML) is a subfield of artificial intelligence, providing systems with the ability to learn from experience and extract knowledge from the available data [1,2,3,4]. It has been widely used in the medical field, from radiology to surgery, from oncology to intensive care [5,6,7,8,9]. Supervised machine learning-based systems have been employed to predict patient deterioration risk [10,11], heart failure onset [12,13], acute kidney injury [14], delirium [15], sepsis [16,17,18,19] and mortality [20,21]. Unsupervised ML, on the other hand, has been used to analyze, cluster and manage large amounts of data that lie beyond clinicians’ ability to handle them. It provides aid to medical staff in the decision-making process, diagnostic method and treatment prescription [22,23]. It can offer support, especially in antibiotic therapy management, although it remains subordinate to international guidelines, protocols and indications [24,25]. In fact, in everyday clinical practice, several issues concerning timing, selection and suitability of antibiotic therapy arise. Recently, the role of ML in infection management and electronic health records has been highlighted [26,27], especially with regard to computerized decision support systems in antimicrobial stewardship [28,29,30,31].

A PubMed database search (at the time of submission) in the field of Intensive Care Units (ICU) with keywords “ICU machine learning” leads to a number of 30 papers just for 2022. Since the year 2019, ML in ICU has been proposed for mortality prediction models from parameters at admission [32], for early warning of sepsis [33,34,35,36,37,38], for early prediction of acute kidney injury (AKI) in pediatric and adult patients [14,39,40,41,42] offering a decision model for the clinician [43,44,45] and for many other specific subfields. ML approaches to vital signs and clinical parameters were used for the comparison of CT findings [46], to personalize levels of care in emergency rooms in mechanically ventilated patients [47,48,49], for prediction of successful extubation [50], and for early prediction of hemodynamic interventions [51,52,53].

Criticisms have been proposed for the reliability of machine learning predictions when the model is not able to generalize to new unforeseen instances, which may originate from a population distant from that of the training population or from an under-represented subpopulation, leading to inaccurate results and consequent decreasing trust of the final users, such as clinicians [54]. Fuzzy logic has been recently proposed for intelligent automated drug administration and therapy [55], for analysis of parameters affecting blood oxygen [56], and for decision making [57]. A warning system for patient classification using fuzzy logic has recently been proposed [58]. The problem of matching clinician expectations with possibilities offered by ML has recently been reviewed, including interpretability vs. explainability and global vs. local explanations [59], and focusing on the broad range of skills that clinicians should develop or refine to be able to fully embrace the opportunities that ML technology will bring in the near future [60].

Focusing on ICU clinicians’ daily work, we propose a ML approach to the problem of patient clustering in ICU, considering fuzzy logic-based categorizing of antibiotic duration of therapy patterns and clinical monitoring observations during ICU stay. It relies on integration of fuzzy logic data normalization and k-mode clustering analysis.

This single-center, retrospective study is to be considered as a proof-of-concept of a tool aimed at guiding clinicians in evaluating ICU patients and improving care and patient outcomes using ML techniques as a side companion to everyday workflow. This would allow future exploration of further opportunities related to the potentiality of our approach, such as exploring specific antibiotic sub-clusters, mortality linked to each cluster and a broader analysis including more variables.

## 2. Materials and Methods

Outline of the project is presented in the Flowchart 1 below.



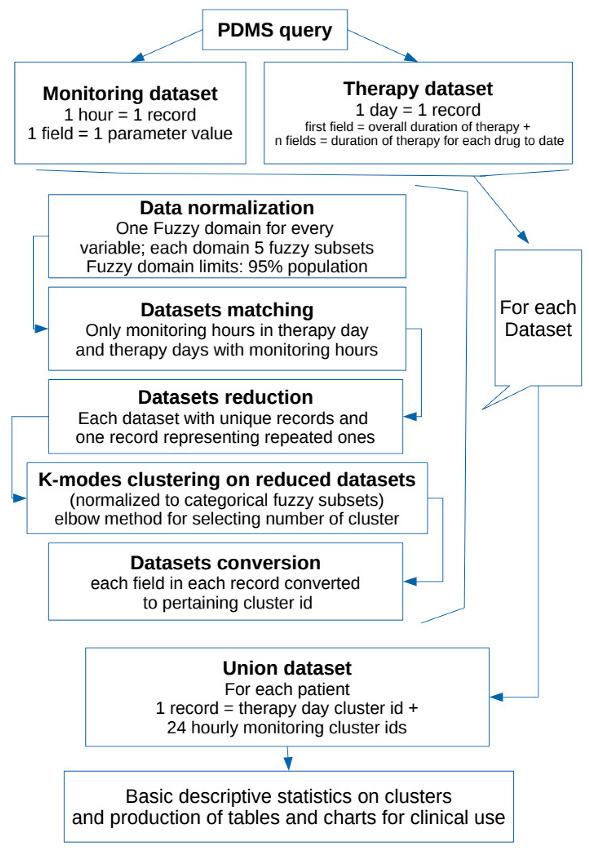



### 2.1. Population and Data Collection

The study was conducted using data pertaining to all patients admitted in the 18-bed ICU at “Fondazione Policlinico Universitario A. Gemelli IRCCS” from 2012–2017.

ICU stay charts (including therapy, fluid balance, and vitals monitoring) are collected, managed, and stored in a patient database management system (PDMS): Digistat by Ascom UMS. All data for this study were extracted and analyzed in anonymized datasets. All clinically validated acquisitions stored in the database were considered for analysis.

For each patient, we assess the duration of antibiotic therapy, consecutive days of treatment and type and combination of antimicrobial agents (each with respective consecutive days of therapy). We decided to define a single “pattern of antimicrobial therapy” as the specific combination in a patient-day (treated as a single record) of the overall duration of therapy and one column for each antimicrobial therapy used in the population during the observation period (resulting in a record of 91 columns having 0 for drugs not used in that day for a patient or the sequential number of therapy days for that drug for a patient).

Duration of therapy was normalized using a fuzzy logic approach to minimize the dispersion effect: duration of therapy (overall and for single antimicrobial agent) for each patient was included in a fuzzy-logic domain having as an output the classification in “none”, “ultra-short”, “short”, “medium”, “long” and “ultra-long” for each item. [58,61,62]

We extracted from our electronic database the following data for all patients (with or without antimicrobial therapy during ICU stay): heart rate, arterial pressure, central venous pressure, intracranial pressure, oxygen saturation, arterial blood gas (ABG), fluid balance (infusion velocity, diuresis and ultrafiltrate from renal replacement therapy if applicable). Each monitoring hour was considered as a whole single record: i.e., for each patient day of antibiotic therapy, we could have N (at least 1 and generally 24 for 24 h) patterns of monitoring parameters (resulting in a record of 46 columns: one for each possible monitoring parameter). Not every patient must necessarily have all the parameters monitored for every hour (i.e., intracranial pressure or continuous renal replacement).

The same normalization approach used for antibiotic therapy was used for each monitoring variable (categories “none”, “very low”, “low”, “medium”, “high”, “very high”).

For each day of antibiotic treatment given to every single patient included, we matched hourly monitoring records and only considered patients with both antimicrobial therapy and monitoring available for analysis.

### 2.2. System Development

Data extracted were analyzed using ad hoc Python scripts (Python 3,6 for Linux). Standard statistical and scientific libraries were used for analysis and graphics (Scipy, Statistics, Sklearn, Numpy, Pandas, Mathplotlib) [63,64]. Fuzzy domains were calculated for all non-null values, assuming data distribution in three overlapping triangular regions and two rectangular trapezoids at the extremities (with the triangular part overlapping the adjacent triangular region). Dimensions of each region and overlapping were automatically calculated for each parameter (starting from basic statistical analysis of the sample), assuming that the limits of the whole domain could describe 95% of population observations. A specific fuzzy domain was produced for every parameter.

### 2.3. Fuzzy Domains and Subsets

The fuzzy domains have 5 subsets each, identified as a verbal description of “very low”, “low”, “medium”, “high”, “very high” (respectively F1, F2, F3, F4, F5 in the charts below) having 4 overlapping regions.

For each parameter (in therapy as well as monitoring set), a specific domain was created starting from the statistical analysis of every non-null data in the population for that parameter. We aimed to obtain a scheme centered on the median value of the distribution (so to better represent all occurrences whose summed frequencies reached at least the 95% of observations). Frequencies of each occurrence of data elements for each parameter were first calculated for all raw observations in the population. Then only the values falling in the range of 95% of observations were extracted and put in a table. Subsequently, the data included in this table of the central 95% of values were used to calculate a new median point, and all values in the population of raw observations outside of this range were automatically assigned to the extreme trapezoid subsets of the domain named “very low” and “very high”. Overlapping regions were set to be equal to one half standard deviation for parameter distribution in the 95% table. Normalization was performed to calculate the membership function of each value (fuzzification: calculating the degree of membership of a value for each subset). The membership function for each value in each domain has a range from 0 to 1: the degree 0 means that the value is not a member of the fuzzy subset; the degree 1 means that the value is fully a member of the fuzzy subset. Values falling in an overlapping region between two subsets were assigned to one of them in the process when their degree of membership was higher than 0.5. A graphic example of a fuzzy domain is presented in Figure 1. 

### 2.4. Datasets

Data of antimicrobial therapy duration and monitoring parameters sets were normalized as previously described. After normalization of each variable, we obtained two normalized datasets (one for antimicrobial therapy and one for monitoring) used for further analysis. Each record of the two datasets has columns in which possible values are only 0 (for drug/monitoring parameter non-present) or 1 to 5 representing fuzzy subsets F1 to F5.

Each record of the two normalized datasets was treated as a potential unique pattern. Two tables (one for antimicrobial therapy and one for monitoring) were created containing only unique patterns (repeated pattern recorded as one in each set, non-repeated pattern recorded as one): assuming each record (i.e., pattern) of the two sets as a single vector (in a Euclidean multi-dimensional space), we matched each of them against all the others in the same set for Euclidean distance. When a distance of 0,0 (no distance) was found we assumed identity.

After this reducing step, we performed cluster analysis using k-modes unsupervised clustering on the two shorter datasets of unique patterns (k-modes is used for clustering categorical variables; it defines clusters based on the number of matching categories between data points—this is in contrast to the more well-known k-means algorithm, which clusters numerical data based on Euclidean distance) [65,66,67].

The number of clusters for every k-modes analysis was calculated by computing the elbow method for each k in a starting range of 25. We decided to use 4 clusters (cost score 285,103) in the therapy dataset and 7 clusters (cost score 41,737) in the monitoring set for k-modes analysis.

Single antibiotic days of therapy and single hours of monitoring (i.e., the records of the two datasets) were then translated as pertaining clusters for each set and so referred to in the remaining part of the analysis (i.e., patient X day 4 h 10 is in cluster 3 in monitoring and in cluster 2 for antibiotic therapy). For each patient a table of days of therapy, marked using therapy pattern cluster id, and 24 columns of consecutive hours of monitoring (each marked as the respective monitoring cluster id as well, or 0 if not present: not all the days of every patient have a complete range of 24 h monitoring) was obtained (i.e.,: patient X day 4, therapy-cluster-id, n = 24 monitoring-cluster-id). Each of the rows in this table, chronologically ordered for each patient, was used for patient ICU stay statistics.

Basic descriptive statistical analysis was performed for each cluster and on ICU stays. Each variable composing a record of the two datasets (Therapy and Monitoring), normalized using fuzzy-logic as described above, is intended as suitable for daily patient evaluation for clinicians alongside results of clustering in tables and charts.

Charts for fuzzy representation of parameters and for clustering were produced, as well as charts for each patient ICU stay (excerpts of 14 h of ICU stay of two random patients are presented in Figures 2 and 3).

## 3. Results

During the period 2012–2017, 5164 ICU admitted patients undergoing antibiotic therapy were extracted from the database. All together, these provided 57,329 patient-day combinations of normalized data, referred to as “antimicrobial therapy pattern”. After removing repeated patterns, the total number of unique antimicrobial therapy patterns was 7763. In the same period, we extracted 935,102 hourly observations of monitoring variables—named “hourly monitoring patterns” (of normalized data)-, referring to patients treated or not with antibiotic therapy, to be assured of including the widest range of measures for each variable. After removing repetitions of patterns, the total number of unique hourly monitoring patterns was 57,966. Considering patients undergoing antimicrobial therapy and available monitoring data, we analyzed 4129 patients and 543,190 h of monitoring in 31,455 patient days.

Duration of antibiotic therapy was normalized according to fuzzy logic in a domain with five fuzzy subsets: “ultra-short”, “short”, “medium”, “long” and “ultra-long” (“none” for non-present data as previously described) for both overall consecutive days of therapy and single drug consecutive days of therapy. Unsupervised cluster analysis collects unique daily antimicrobial therapy patterns in four clusters. Table 1 shows frequencies of every fuzzy domain subset in each antibiotic therapy cluster.

Each monitoring variable has been fuzzy normalized in five fuzzy subsets: “very low”, “low”, “medium”, “high”, “very high” (“none” for non-present data). Unsupervised cluster analysis collects unique hourly parameter monitoring patterns in seven clusters. Frequencies of every cluster in each fuzzy domain of hourly monitoring pattern are shown in Table 2.

Table 3 shows an example of a fuzzification method applied to the parameter of total days of consecutive antimicrobial therapy.

Figure 1 represents an example of a fuzzy domain.

Figure 2 and Figure 3 show graphic representations of an example of 14 monitoring hours and changes in antimicrobial therapy cluster according to monitoring cluster.

Table 4 shows the contingency table between antimicrobial therapy pattern clusters and monitoring pattern clusters. (Therapy clusters are named from T0 to T3; monitoring clusters from M0 to M6). 

Considering each single patient ICU stay, we found that the maximum number of antimicrobial therapy clusters per ICU stay was four. We observed groups of patients having one, two, three, or four different therapy clusters during stay, not saying which of the therapy clusters—Nabtcl... Table 5 presents a synoptical view of therapy clusters, monitoring clusters and consecutive days of therapy stratified by number of therapy clusters in ICU stay (Nabtcl). 

For more details on k-modes cluster analysis compared to k-means cluster analysis on the same data, see Appendix A.

## 4. Discussion

In their 2019 paper, Al-Dmour et al. [59] proposed an implementation of a warning system that utilizes fuzzy logic for patient classification. Starting from the literature reported in the paper, they proposed that, when the diagnosis might not be very clear, and there is an element of uncertainty in the reasoning process followed to reach a diagnosis, the use of fuzzy logic has shown to provide an effective reasoning methodology that can address uncertainties and vagueness since it allows for the representation of imprecise knowledge. They used parameters coming from assessment of the Modified Early Warning Scoring (MEWS) system (physiological parameters or vital signs readings such as temperature, blood pressure, heart rate, respiratory rate, oxygen saturation, and blood sugar) [66]. MEWS score ranges, in the cited paper, for each parameter were reviewed by experts and transferred to linguistic terms or categories, each category identifies a fuzzy set; a membership function for each set and a range were defined for each function.

In the present paper, we used a similar fuzzy logic-based approach for data translation. We used a statistical method instead of expert definition for fuzzy domain range settings; we think that this method could be useful for continuous retraining of the model or for different kinds of applications (i.e., for a single parameter performing stratification on a subset of the general population or focusing only on a single patient during the ICU stay). In the present paper, the typical process of a Fuzzy Inference System of mapping an input to an output is used only for the first stages: defining the membership functions and converting the data to linguistic categories. We applied fuzzy logic conversion to linguistic terms to both ICU patient monitoring parameters and antimicrobial therapy expressed in days. We then used converted parameters for exploration of clustering methods.

In our approach, we do not define a priori conditions for defining clusters either in antimicrobial therapy or monitoring. The assumption is: we are not looking only at a specific antibiotic (or combination of multiple drugs), its duration of therapy in patients with specific severity score (i.e., MEWS), or for a specific clinical condition (as in the papers of Bloch et al., Pettinati et al., Mao et al., Fleuren et al., Hou et al., Wang et al. [33,34,35,36,37,38]. We do not, at the moment, even imply an association with outcome or length of stay in ICU to therapy patterns and clusters. We do not want to suggest therapy changes using specific patterns of empirical therapy outside of protocols when clinical modifications are observed. In other words: we do not propose a rule-based engine for prediction, but rather we show a model for data presentation to clinicians.

Our definition of therapy pattern is based on duration of therapy (overall and for each drug during ICU stay: i.e., patient X is at day 7 of antimicrobial therapy, receiving drug Y for 3 days and drug Z for 7 days). Clusters of this interpretation of therapy patterns were analyzed. We can say, looking at the frequency distribution of fuzzy subsets in each therapy cluster, that cluster 1 and 2 (T1 and T2 in tables and charts) contain more fuzzy subsets F1 (11% in T1; 21% in T2) and F2 (49% in T1 and 47% in T2) (representing “very low” and “low” duration of therapy), and that in cluster 3 (T3) there are more of summed “high” and “very high” fuzzy subsets (F4 and F5 = 36%), meaning long and very long duration of therapy (but also 34% of F2 = “low” duration). Cluster T0 has a very low percentage of “long” and “very long” duration of therapy (F4 = 9%, F5 = 7%).

This does not mean that a patient starting antimicrobial therapy (at day 1) is per se in cluster 1 (T1), but if he/she is in cluster T1 he/she shares something (maybe the drugs used in the pattern or the combination of overall duration of therapy and the same drugs) with other patients. With passing days of therapy, the same patient could change his/her pattern and the day of therapy could be in a different cluster. The new cluster he/she is in, means he/she shares the condition of overall and single/multiple drugs duration of therapy with other patients.

In our population, the maximum number of therapy clusters per patient ICU stay was four (i.e., there were patients who changed therapy pattern and overall duration of therapy to a maximum of four different clusters). Patients with only one therapy cluster during ICU stay had therapy days mostly in cluster T0; patients with three clusters in the ICU stay had equal frequencies of the four therapy clusters and patients with four therapy clusters in the same ICU stay most frequently had cluster 1 (40%).

The same point of view must be used for monitoring. Monitoring clusters 0 and 3 (M0 and M3 in tables and charts) have most of summed fuzzy subsets F4 and F5 (49% in M0 and 38% in M3), while M1 and M6 have more of summed F1 and F2 (M1 47%; M6 51%) than the others. The severity of having “very low” or “very high” for a parameter is parameter-dependent (“very high” Heart rate and “very high” SpO2 are different). Therefore, assuming that a monitoring hour in cluster M0 is better than M6 is erroneous. However, small (and not so small) variations for monitoring parameters are compensated using fuzzy subsets and clustering. Medical evaluation of single variables (even if expressed in descriptive terms) and their association is always needed. If a patient changes his/her cluster for monitoring from one hour to another, considerations of similarities to other patients should be made (the patient stopped being equal to him/herself).

Combining the two sets (antimicrobial therapy and monitoring), we can obtain a broader space for medical discussion. Again not a priori, but rather a patient-centered discussion nearer to precision-medicine.

In our population, we observed relatively higher frequencies of monitoring cluster M4 in all therapy clusters. The contingency table is presented in Table 4.

Patient profiling is a physician/human attitude, often addressed as a clinician’s experience. In everyday ICU practice, data (lab, vitals, imaging, and physical examination) are summarized in briefings and hand-over using more descriptive qualitative rather than quantitative expressions. Providing a method to measure that clinical experience, a proxy for that automatic *gestalt* evaluation that an experienced clinician sometimes effortlessly, but often only after long, hard consideration and consultation with colleagues, relies upon for decision making, is what we wanted to achieve with the present work.

Critical patients are so defined as a “whole” and clustering is strictly implicit in the definitions. Scoring systems for severity are daily used for staking patients in stratifications often based on comorbidity, reasons for ICU stay, chronicity, etc. Smart alarms based on variations in vitals are used in monitoring equipment. Using scoring systems based on fixed ranges for vitals instead of five different grades could limit the prospectical vision of a patient deterioration.

We explored the possibility of combining medical reasoning and language (expressions like: “worsening”, “stable”, “always the same”, “similar to that kind of patient”, etc.) to the machine learning world.

Using fuzzy logic-style normalization for data and performing cluster analysis on these sets could be seen as choosing different keys when trying to open a lock: the shape of the key can be similar to some but very different to others. Exploring similarities can be an option, but having evidence of differences could be more informative, particularly when changes occur in the same patient.

We decided to use data from our database instead of public domain datasets for vitals and therapy. In our ICU, antimicrobial therapy has always been prescribed using a stewardship approach. Changes in therapy are (and have been) always made, in our ICU, using both protocols and patient evidence (laboratory findings, microbiological evidence, patient risk factors). Nonetheless, changes in patient conditions can lead to diagnostic and therapy changes using a human (experience) sense of probability more than weighted or statistical evidence.

Cluster analysis of antimicrobial therapy and monitoring as separate entities is not what we (as intensivist care specialists) call “the real world”. We decided, therefore, to combine two sets in one presentation to reproduce what we consider in single-patient analysis when antimicrobial therapy changes are concerned (i.e., How long has he been on therapy? How long on XXX-drug? And in combination? How is he doing with his hemodynamics/temperature/renal-organ-functions? Never did better/worse?).

As proof of this concept, we wanted to develop a system that could provide the clinicians a synoptical view of data as a support for therapy evaluation. We, then, decided to have an unsupervised cluster analysis on therapy and monitoring sets. The monitoring and therapy patient data are expressed in a descriptive/categorical form, typical of the human language.

As evident, our proposal for antimicrobial therapy evaluation using temporal categories cannot substitute general/local protocols or recommendations. The cluster analysis of the datasets, and the combination of these, can provide statistics for clinical discussion about patients. Suppose a patient is in a cluster of antimicrobial therapy (i.e., overall days of therapy and each drug consecutive day of therapy). In that case, he/she shares a statistical nearness with other patients independently from reasons of ICU stays or severity score at admission (i.e., Apache-II or SAPS-II). In a retrospective analysis, this is always a subtly implied entity. Using experience and weighing the evidence of shifting from one cluster to another can give an idea in an ongoing view of what the missing data might be (antimicrobial evidence, vitals changing for not-yet-known new illness, and so forth).

What we yielded, and propose here, is an embryonal series of graphic presentations for each patient’s progress during his/her ICU stay based on the fuzzy translation of each parameter and cluster linkage for clinical discussion. In our mind, this should be an integrated tool of our digital patient record as a natural companion to the numeric tables for vitals and therapy charts.

Future research opportunities using this approach could be: analysis of whole patient therapy (not only antimicrobial); comparison of patients within same clusters and co-morbidities; exploration of antimicrobial therapy clusters for severity and mortality focusing on use of specific antibiotic patterns; temporal analysis of antimicrobial drug use.

As criticism about the reasons why clinician trust for ML methods were proposed [60], we hope that our approach could improve trust in ML methods.

This study has some limits. It is a single center retrospective study based on ICU patients. Monitoring data and therapy are limited to a short set. It is an embryonal exercise proposed as a proof of concept of a methodology. Upon these premises, no clinical validation, nor prediction model is proposed. Antimicrobial therapy was used, but concurring drugs/therapies were not considered. We did not perform any kind of analysis on subsets of antimicrobial drugs or combinations of them (i.e., different classes use, daily drug dosage, etc.). We did not construct a fully structured fuzzy-logic inference engine, using only conversion to linguistic terms of the variables: no expert revision and validation, nor comparison to existing severity scores, of the fuzzy domain ranges has been performed. We performed clustering analysis just using k-modes algorithm: in Appendix A, a raw comparison to k-means analysis on the same datasets is only proposed as an example, not as demonstration of robustness of our approach. Considering an expansion of the datasets (especially for therapy) could lead to different clustering and need for a wider population or change in the methods proposed here.

## 5. Conclusions

A practical clinical approach to the constant increase in patient data flow in a modern ICU is a widespread issue. Antimicrobial therapy is an issue often discussed during patient examination, briefing or handover. Consideration of a patient’s modifications in terms of a group of parameters, normalized using fuzzy categories, could be used as a logical map for describing his/her ICU stay. Far from being conclusive in considering antimicrobial therapy prescription in intensive care, our approach could be a guide for daily clinical patient management for diagnostics and care. Extending the same methods to other patient parameters (i.e., non-antimicrobial therapy, laboratory findings, microbiological evidence, etc.) collected during the ICU stay and presenting them to the intensivist could be, as a further extension of the present paper, useful to paint a more precise picture of the clinical patient condition when modifications are observed.

Using fuzzy sets and clustering could be, in our opinion, a starting point for a different approach to presenting ML prediction algorithms to the clinician for precision medicine.

## Figures and Tables

**Figure 1 antibiotics-11-00304-f001:**
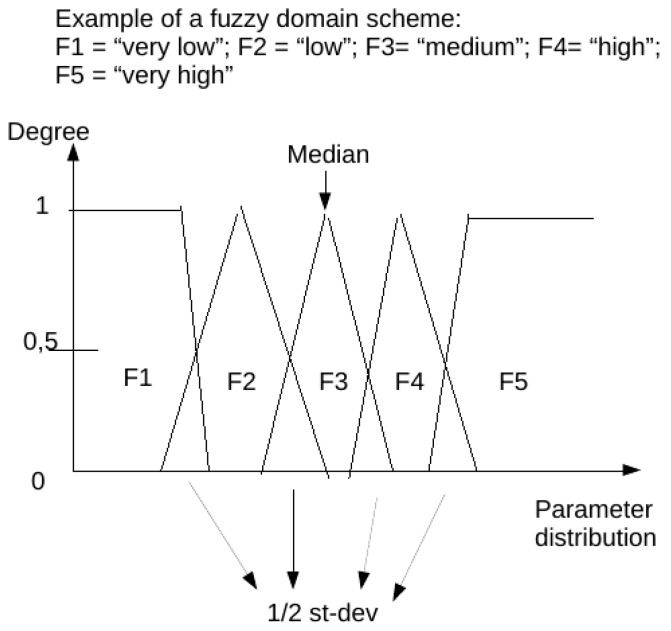
Example of a fuzzy domain. Fuzzy subsets (named from F1 to F5) are represented as geometrical graphical charts.

**Figure 2 antibiotics-11-00304-f002:**
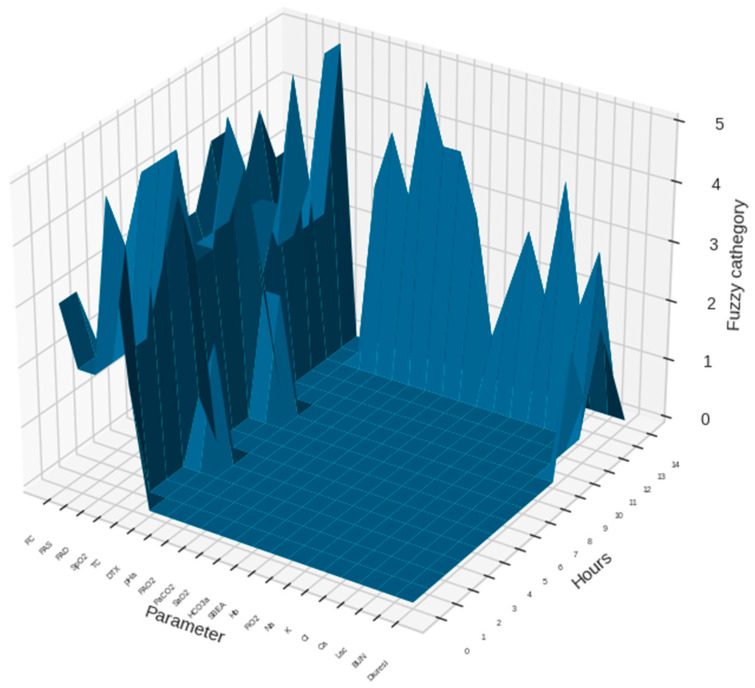
An example of 14 monitoring hours with a sample of the monitoring parameters in a patient with normalized (fuzzified) data is presented. The dimensional view gives information on the data flow for each parameter during the observation period. Height of the curves in the chart are in units representing the normalized values for each parameter.

**Figure 3 antibiotics-11-00304-f003:**
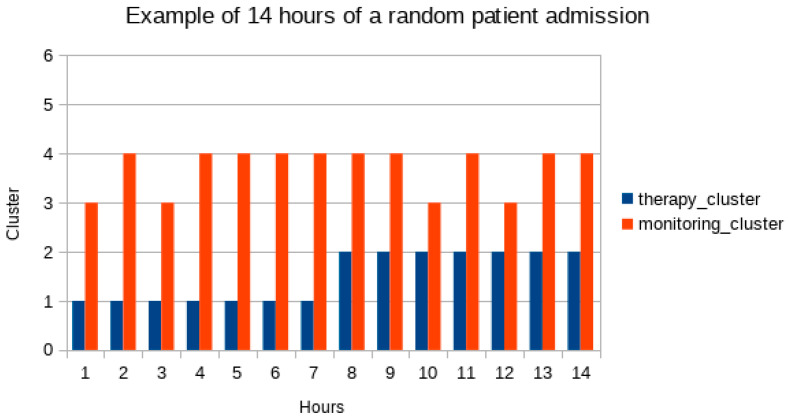
Changes in antimicrobial therapy clusters in a random patient in the population. Respective monitoring clusters in the same hour are charted. In the sample of 14 h here proposed the patient is in therapy cluster 1 (blue bars) for 7 h and starting at hour 8 is in cluster 2. Next to blue bars are red bars for monitoring clusters in the same hours.

**Table 1 antibiotics-11-00304-t001:** Clustering of unique daily antimicrobial therapy patterns: distribution of frequencies of fuzzy domain subsets per antibiotic therapy cluster.

Cluster	N Patterns	Non-Null	F1	F2	F3	F4	F5
T0	3551	10,324	0.24	0.38	0.22	0.09	0.07
T1	1159	4437	0.11	0.49	0.10	0.11	0.20
T2	1569	4257	0.21	0.47	0.13	0.10	0.09
T3	1484	5739	0.16	0.34	0.14	0.14	0.22

Clusters of therapy are named from T0 to T3, fuzzy categories from F1 to F5. N-Patterns: number of unique antimicrobial day patterns per cluster. Non-null: Total number of non-null fuzzy subsets in the patterns. Chi-square value 1818.4; *p* < 0.05.

**Table 2 antibiotics-11-00304-t002:** Clustering on unique hourly parameter monitoring patterns: distribution of frequencies of sum of fuzzy subsets per monitoring cluster.

Cluster	N	Non-Null	F1	F2	F3	F4	F5
M0	11,520	80,158	0.07	0.17	0.27	0.34	0.14
M1	10,017	73,525	0.09	0.38	0.34	0.10	0.10
M2	14,624	93,037	0.10	0.27	0.35	0.14	0.14
M3	7717	49,562	0.08	0.20	0.33	0.30	0.08
M4	2391	16,732	0.09	0.34	0.28	0.25	0.05
M5	3358	22,431	0.07	0.32	0.24	0.24	0.13
M6	8339	54,657	0.12	0.39	027	0.10	0.12

Monitoring clusters are named from M0 to M6, fuzzy categories from F1 to F5. N: unique hourly parameter patterns per cluster. Non-null: Total of non-null fuzzy subsets in the patterns. Chi-square value 35,848.5; *p* < 0.05.

**Table 3 antibiotics-11-00304-t003:** Basic statistics for total days of antimicrobial therapy set (considering only observations reaching 95% in frequencies corresponding at day 22 for this parameter).

	Mean	Minimum		Maximum	Observed	Median	Range	Standard Deviation	
**Days**	6.43	0		22	57329	11	22	5.7	
**Very low (F1)**		**Low (F2)**		**Medium (F3)**		**High (F4)**		**Very high (F5)**	
**VLL**	**VLH**	**LL**	**LH**	**NL**	**NH**	**HL**	**HH**	**VHL**	**VHH**
−1.8	3.6	1.8	9.1	7.3	14.6	12.8	20.2	18.3	23.8

Limits in the fuzzy domain for total days of therapy (limits are in days). Values equal or over the limit of VHH observation are classified in category (subset) 5; equal or below VLL observation are classified in category (subset) 1.

**Table 4 antibiotics-11-00304-t004:** Contingency table (frequencies) in the population.

	M0	M1	M2	M3	M4	M5	M6
T0	0.07	0.05	0.16	0.04	0.01	0.01	0.10
T1	0.04	0.02	0.07	0.02	0.00	0.01	0.04
T2	0.04	0.03	0.07	0.02	0.01	0.01	0.04
T3	0.03	0.02	0.05	0.02	0.00	0.01	0.03

X = monitoring cluster (M0 to M6); Y = antimicrobial therapy cluster (T0 to T3). Chi-square Value 2368.3; *p* < 0.05.

**Table 5 antibiotics-11-00304-t005:** Synoptical view of therapy clusters, monitoring clusters and consecutive days of therapy stratified by number of therapy clusters in ICU stay (Nabtcl).

	Nabtcl 1	Nabtcl 2	Nabtcl 3	Nabtcl 4					
**N ICU stays (Total 4129)**	2377	1414	306	32					
**N hours (Total 543,190)**	137,667	276,361	113,197	15,965					
**Therapy clusters per ICU stay**					**Fuzzy subsets**				
**Therapy cluster**	**Nabtcl 1**	**Nabtcl 2**	**Nabtcl 3**	**Nabtcl 4**	**F1**	**F2**	**F3**	**F4**	**F5**
T0	0.78	0.38	0.25	0.12	0.24	0.38	0.22	0.09	0.07
T1	0.00	0.24	0.28	0.40	0.11	0.49	0.10	0.11	0.20
T2	0.13	0.22	0.24	0.28	0.21	0.47	0.13	0.10	0.09
T3	0.09	0.16	0.23	0.21	0.16	0.34	0.14	0.14	0.22
Chi-square 100,641.5; *p* < 0.05									
**Monitoring clusters per ICU stay**					**Fuzzy subsets**				
**Monitoring cluster**	**Nabtcl 1**	**Nabtcl 2**	**Nabtcl 3**	**Nabtcl 4**	**F1**	**F2**	**F3**	**F4**	**F5**
M0	0.16	0.16	0.18	0.23	0.07	0.17	0.27	0.34	0.14
M1	0.12	0.13	0.12	0.12	0.09	0.38	0.34	0.10	0.10
M2	0.35	0.34	0.34	0.31	0.10	0.27	0.35	0.14	0.14
M3	0.09	0.10	0.10	0.09	0.08	0.20	0.33	0.30	0.08
M4	0.03	0.03	0.04	0.02	0.09	0.34	0.28	0.25	0.05
M5	0.03	0.03	0.04	0.04	0.07	0.32	0.24	0.24	0.13
M6	0.22	0.21	0.19	0.19	0.12	0.39	0.27	0.10	0.12
Chi-square 1918.8; *p* <0.05									
**Overall Consecutive Days of therapy as fuzzy subsets**									
**Fuzzy subset**	**Nabtcl 1**	**Nabtcl 2**	**Nabtcl 3**	**Nabtcl 4**					
F1	0.28	0.14	0.07	0.06					
F2	0.37	0.38	0.25	0.22					
F3	0.13	0.16	0.17	0.16					
F4	0.08	0.11	0.15	0.17					
F5	0.14	0.22	0.35	0.39					
Chi-square 40,867.1; *p* < 0.05									

For each therapy and monitoring cluster frequency of fuzzy subset of the pertaining cluster are presented (view by row). Overall consecutive days of therapy are presented for each group of ICU stays as fuzzy subsets. Therapy clusters are T0 to T3; Monitoring Clusters are M0 to M6; Fuzzy subsets are F1 to F5.

## Data Availability

The data presented in this study are available upon a reasonable request from the corresponding author.

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
