# Peer review of "Machine Learning and Antibiotic Management"

_antibiotics, 2022, doi:10.3390/antibiotics11030304_

Round 1
Reviewer 1 Report
The authors applied fuzzy logic and k-modes unsupervised clustering techniques for the patients in ICU to analyze the antimicrobial therapy and monitoring patterns. The outcomes with interpretation were explained in the manuscript. Please see below my comments.
- The authors should describe relevant works and compare their work with them.
- Although the performance of their methods was highlighted in the manuscript, they should validate the results.
- The manuscript should include possible future research opportunities.
- If possible, please make the data and scripts publicly available for the research communities.
Author Response
The authors applied fuzzy logic and k-modes unsupervised clustering techniques for the patients in ICU to analyze the antimicrobial therapy and monitoring patterns. The outcomes with interpretation were explained in the manuscript. Please see below my comments.
- The authors should describe relevant works and compare their work with them.
- Although the performance of their methods was highlighted in the manuscript, they should validate the results.
- The manuscript should include possible future research opportunities.
- If possible, please make the data and scripts publicly available for the research communities.
Response to reviewer:
Respected Reviewer,
Thank you for taking the time to review and comment on our manuscript, “Machine learning and antibiotic management”. We found your suggestions constructive and we included them in the revised version.
Point 1. We have included comparisons to previous relevant works as well as future opportunities , as suggested.
Point 2. This study is to be considered a presentation of an approach, useful for clinical patient evaluation; validation (and more critically, clinical validation) is possible, and we intend to attempt this in a subsequent study.
Point 3. We included possible future research emerging from the present paper at the end of the discussion.
Point 4. As requested by the reviewer, datasets can be made available upon a reasonable request and after authorization by our institution, while scripts can be obtained from the authors.
Furthermore, a native English-speaker corrected the revised manuscript before submission.
Thank you again for your thoughtful comments.
Sincerely,
Riccardo Maviglia on behalf of all the co-authors
Reviewer 2 Report
Please please revise the english: Italian expression does not traslate in most cases literally to english and this can make the paper too hard to read for a non italian reader.
Abstract is not sufficently expressive of the contents of the paper and on the other hand Introduction reports standard basic notions about ML and fuzzy reasoning and could be compressed.
The fuzzy clustering approach is performed soundly.
Membership functions have been selected with due care and according to the recognized state of the art in fuzzy reasoning.
The problem with this report of a research that looks quite at its begin is its aim.
The "exercise" of sorting and cathegorizing data using a standard fuzzy logic approach is correct, but does really give insights to the clinicians about their approach to anti-biotic therapy? This is the main weakness of this paper.
More technically:
a) using the knee method for proper clustering it is not an absolute criterion and specific domain arguments must be given to judge the chosen number of cluster suitable for the data.
b) The paper would be better if some other algorithm (hard K-means, or DBSCAN) is performed as well and results are compared.
c) Euclidean distance as dissimilarity measure must be justified. Do the data have similar ranges or have they been normalized?
Author Response
Reviewer 2.
Please please revise the english: Italian expression does not traslate in most cases literally to english and this can make the paper too hard to read for a non italian reader.
Abstract is not sufficently expressive of the contents of the paper and on the other hand Introduction reports standard basic notions about ML and fuzzy reasoning and could be compressed.
The fuzzy clustering approach is performed soundly.
Membership functions have been selected with due care and according to the recognized state of the art in fuzzy reasoning.
The problem with this report of a research that looks quite at its begin is its aim.
The "exercise" of sorting and cathegorizing data using a standard fuzzy logic approach is correct, but does really give insights to the clinicians about their approach to anti-biotic therapy? This is the main weakness of this paper.
More technically:
a) using the knee method for proper clustering it is not an absolute criterion and specific domain arguments must be given to judge the chosen number of cluster suitable for the data.
b) The paper would be better if some other algorithm (hard K-means, or DBSCAN) is performed as well and results are compared.
c) Euclidean distance as dissimilarity measure must be justified. Do the data have similar ranges or have they been normalized?
Response to reviewer:
Respected Reviewer,
Thank you for taking the time to review and comment on our manuscript entitled “Machine learning and antibiotic management”. We found your suggestions constructive and we included them in the revised version. We respond to each comment individually below.
- We have corrected grammatical and spelling mistakes. Furthermore, a native English speaker revised the paper before new submission.
- “Abstract” and “Introduction” have been re-drafted, according to your suggestions, in order to highlight the contents of the paper, limiting unnecessary notions.
- Our study probably does look quite at its beginning, as it has an explorative aim. It is to be considered as a proof of concept, an “exercise” to explore a different approach in ICU patient management, integrating antibiotic therapy and vital parameters. We have highlighted opportunities and the potentiality of this tool in the main text.
- The reviewer doubts if our paper really gives clinicians insights into their approach to antibiotic therapy. We think that, when a critical patient shows signs of deterioration, the guidelines (either local or general protocols) for clinical evaluation of empirical antibiotic therapy should be considered first. The most difficult part of the decision is the evaluation: the review of the patient’s data from the time of admission (including but not exclusively antimicrobial therapy) and the trends that can be seen in their monitoring and therapy. A view that could summarize and better represent patient history giving a statistical basis to human observation and intuition could be quite helpful in our hypothesis. However, we do not present a clinical validation in this paper, but rather intend to undertake such an endeavor in a future study. Taking the wise advice of the reviewer into good account on this point, we have stressed the limits of our approach in the new draft.
- The knee method is not an absolute criterion, but is commonly used and presented as acceptable in literature. We provide arguments for the chosen number of clusters in an appendix to the revised paper.
- We performed K-means clustering as well, but preferred to present the K-mode approach as it could raise less criticism about its use for non-continuous variables (normalized data using fuzzy categories). We provide comparison with k-means clustering as supplementary material in the appendix.
- Euclidean distance was used in the presented work only for normalized data. Each parameter composing a record was normalized (using Fuzzy approach as described) and records were then compared using Euclidean distance before clustering.
Thank you again for your thoughtful comments.
Sincerely,
Riccardo Maviglia on behalf of all the co-authors
Reviewer 3 Report
This paper studies Machine learning and antibiotic management There are a few weaknesses that should be addressed in this paper. This can provide an additional tool for assessing the progress of complex patients. The use
of fuzzy logic normalization of parameters can be useful as a guide in briefings and in handover or for patient and family communications as it is nearer to human language than raw numbers. This version is not acceptable at all. Therefore, I strongly suggest the rejection of this paper. My suggestions are as follows:
General comments:
- As the first step, I strongly suggest that the paper be proofread and reread meticulously again, particularly in regard to the spelling and grammatical mistakes.
- The paper should be revised to include at least 20 recent references on Machine learning in healthcare, in particular 2021 and 2022.
- Your introduction is too short. It is necessary to include additional information in the introduction part.
- Please mention the structure of your paper at the end of the introduction part.
- It would be so helpful If you consider a flowchart at the end of the introduction as well.
- The limits of the results obtained in this paper are not mentioned. This point should be explained.
- Comparisons with existing approaches are missing.
- Your tables structure is so weird and irregular.
- What is the purpose of the statistical analysis?
- Figure 2 is not visible.
Other comments:
- In line 32 you mentioned:
"Machine learning (ML) has been playing a pivotal role over the past few years."
Please ML summery in line 15 when you mentioned:
"Machine learning and cluster analysis applied to the clinical setting"
2.In lines 46-50 you mentioned:
"It provides aid to medical staff in the decision-making process, diagnostic method and treatment. It can offer support, especially in antibiotic therapy management, relies despite relying on international guidelines, protocols and indications. It’s a tool not to give simple answers, but rather to induce clinicians to make further questions, even simply making them aware of upcoming issues and disease progression in advance"
It is too long with a weird explanation. What is your mean by mentioning "It’s a tool not to give simple answers, but rather to induce clinicians to make further questions, even simply making them aware of upcoming issues and disease progression in advance"
3. You have considered 16 references in one place in lines 70-71 as follows:
"Duration of therapy was normalized using a fuzzy logic approach to minimize dispersion effect: duration of therapy (either overall and for single antimicrobial agent) for each patient was included in a fuzzy-logic domain having as an output the classification in “none”, “ultra-short”, “short”, “medium”, “long” and “ultra-long” for each item. [12- 70
27]"
Considering 15 references without any clarification is the main reason for my strong suggestion to reject this paper.
Author Response
This paper studies Machine learning and antibiotic management There are a few weaknesses that should be addressed in this paper. This can provide an additional tool for assessing the progress of complex patients. The use
of fuzzy logic normalization of parameters can be useful as a guide in briefings and in handover or for patient and family communications as it is nearer to human language than raw numbers. This version is not acceptable at all. Therefore, I strongly suggest the rejection of this paper. My suggestions are as follows:
General comments:
- As the first step, I strongly suggest that the paper be proofread and reread meticulously again, particularly in regard to the spelling and grammatical mistakes.
- The paper should be revised to include at least 20 recent references on Machine learning in healthcare, in particular 2021 and 2022.
- Your introduction is too short. It is necessary to include additional information in the introduction part.
- Please mention the structure of your paper at the end of the introduction part.
- It would be so helpful If you consider a flowchart at the end of the introduction as well.
- The limits of the results obtained in this paper are not mentioned. This point should be explained.
- Comparisons with existing approaches are missing.
- Your tables structure is so weird and irregular.
- What is the purpose of the statistical analysis?
- Figure 2 is not visible.
Other comments:
- In line 32 you mentioned:
"Machine learning (ML) has been playing a pivotal role over the past few years."
Please ML summery in line 15 when you mentioned:
"Machine learning and cluster analysis applied to the clinical setting"
2.In lines 46-50 you mentioned:
"It provides aid to medical staff in the decision-making process, diagnostic method and treatment. It can offer support, especially in antibiotic therapy management, relies despite relying on international guidelines, protocols and indications. It’s a tool not to give simple answers, but rather to induce clinicians to make further questions, even simply making them aware of upcoming issues and disease progression in advance"
It is too long with a weird explanation. What is your mean by mentioning "It’s a tool not to give simple answers, but rather to induce clinicians to make further questions, even simply making them aware of upcoming issues and disease progression in advance"
- You have considered 16 references in one place in lines 70-71 as follows:
"Duration of therapy was normalized using a fuzzy logic approach to minimize dispersion effect: duration of therapy (either overall and for single antimicrobial agent) for each patient was included in a fuzzy-logic domain having as an output the classification in “none”, “ultra-short”, “short”, “medium”, “long” and “ultra-long” for each item. [12- 70- 27]"
Considering 15 references without any clarification is the main reason for my strong suggestion to reject this paper.
----
Respected Reviewer,
Thank you for taking the time to review and comment on our manuscript, entitled “Machine learning and antibiotic management”. We found your suggestions constructive and we included them in the revised version. We respond to each comment individually below.
- We have corrected grammatical and spelling mistakes. Furthermore, a native English speaker revised the paper before new submission.
- According to your advice, we have performed an additional review of more recent papers in order to report further and more recent references.
- The “Introduction” has been re-drafted, as per your suggestions, in order to highlight the contents and the structure of the paper, limiting unnecessary notions. As you suggested, we added a flowchart illustrating the methods at the end of the introduction to provide a clearer vision of the project.
- A paragraph concerning limits of the study has been added.
- Comparisons to other methods fall outside the scope of our research, which is to be considered as a presentation of a possible approach with an explorative aim. Nevertheless, wherever possible, we added comparisons in the discussion.
- Table structure has been redrafted, as you suggested.
- With regards to statistical analysis, we performed only basic statistical analysis to describe clusters and groups.
- We have resolved issues related to figure 2.
- We have included abbreviations at first mentioned in the text and noted parenthetically after expansion.
- We have removed weird and unclear sentences from the redrafted introduction.
- We have clarified and explicated references to other papers in the whole draft, avoiding the combination of too many references together
Thank you again for your thoughtful comments.
Sincerely,
Riccardo Maviglia on behalf of all the co-author
Round 2
Reviewer 2 Report
MOst of the requests done on previuos version of the paper have been done. The paper is now much more readable and the purporse of this research (that is orginal but at its early development) is now much more clear.
Methodological weak points have been clarified in the new version of the paper as well in the additional file sprovided to this referee.
Reviewer 3 Report
This is version is available for the publication.